# Advances in the Involvement of Metals and Metalloids in Plant Defense Response to External Stress

**DOI:** 10.3390/plants13020313

**Published:** 2024-01-20

**Authors:** Lingxiao Zhang, Zhengyan Liu, Yun Song, Junkang Sui, Xuewen Hua

**Affiliations:** 1School of Agricultural Science and Engineering, Liaocheng University, Liaocheng 252000, China; 2320190315@lcu.edu.cn (Z.L.); suijunkang@lcu.edu.cn (J.S.); 2School of Life Sciences, Liaocheng University, Liaocheng 252000, China; songyun@lcu.edu.cn

**Keywords:** metal(loid), element defense hypothesis, abiotic stress, biotic stress, nanoparticles

## Abstract

Plants, as sessile organisms, uptake nutrients from the soil. Throughout their whole life cycle, they confront various external biotic and abiotic threats, encompassing harmful element toxicity, pathogen infection, and herbivore attack, posing risks to plant growth and production. Plants have evolved multifaceted mechanisms to cope with exogenous stress. The element defense hypothesis (EDH) theory elucidates that plants employ elements within their tissues to withstand various natural enemies. Notably, essential and non-essential trace metals and metalloids have been identified as active participants in plant defense mechanisms, especially in nanoparticle form. In this review, we compiled and synthetized recent advancements and robust evidence regarding the involvement of trace metals and metalloids in plant element defense against external stresses that include biotic stressors (such as drought, salinity, and heavy metal toxicity) and abiotic environmental stressors (such as pathogen invasion and herbivore attack). We discuss the mechanisms underlying the metals and metalloids involved in plant defense enhancement from physiological, biochemical, and molecular perspectives. By consolidating this information, this review enhances our understanding of how metals and metalloids contribute to plant element defense. Drawing on the current advances in plant elemental defense, we propose an application prospect of metals and metalloids in agricultural products to solve current issues, including soil pollution and production, for the sustainable development of agriculture. Although the studies focused on plant elemental defense have advanced, the precise mechanism under the plant defense response still needs further investigation.

## 1. Introduction

Soil-grown plants, as sessile organisms, keep coordinating external environmental conditions, which may affect their growth and development detrimentally throughout every stage of their life cycle. As primary producers, plants extract mineral nutrients from the soil through various pathways, employing optimized uptake and utilization strategies to fulfill their nutritional requirements [1]. The majority of these nutrient elements are indispensable for plant growth, participating in the composition of plant organs, metabolic functions, and their ability to respond to stress. The composition and bioavailability of nutrient elements are influenced by soil and environmental conditions that may cause plant disorders [2]. External stresses are typically divided into two categories: biotic and abiotic stresses. Biotic stresses involve challenges like herbivore consumption and pathogen invasions by fungi, bacteria, and viruses, while abiotic stresses encompass conditions such as drought, salinity, heat, and heavy metal toxicity [3,4]. These stresses can jeopardize plant growth across various phases, from seed germination to nutrient utilization, overall plant physiology, distribution, and productivity [5]. Furthermore, adverse stressors are exacerbated according to global climate change and soil contamination scenarios [6,7]. As stationary receptors, plants can sense and perceive signals from external stressors, and then they can activate signaling mechanisms to resist stress [8]. Over the course of evolution, both root and aerial tissues have developed sophisticated mechanisms at physiological, biochemical, and molecular levels to combat these adverse effects [9]. For instance, plants may restructure their root architecture to adapt to drought stress.

The element defense hypothesis (EDH) was raised years ago, suggesting that plants uptake elements as a defense mechanism against pathogen attack and herbivory consumption [10]. Subsequent evidence strongly supported the idea of elements playing a beneficial role in how plants respond to external stress [11,12,13]. Among the advances of EDH, trace metals and metalloids are attractive, playing important roles in plant growth and metabolism. Numerous trace metals and metalloids (semimetals) are absorbed by plants through specific transporters for their utilization. Preliminary studies predominantly support the EDH in hyperaccumulating plants, recognized for their exceptional ability to accumulate heavy metals. These extraordinary plants exhibit the capacity to accumulate heavy metals thousands of times higher than normal plants, allowing them to thrive in contaminated soil and survive in adverse conditions [14]. The database of hyperaccumulators currently catalogs 721 species demonstrating hyperaccumulation of metals and metalloids, with some species even displaying the capacity to hyperaccumulate more than one metal [15]. Extraordinary high accumulation and tolerance of hyperaccumulators to various metals and metalloids make them versatile tools for phytoremediation [16]. Hyperaccumulation of metals and metalloids in plant tissues can elucidate the adaptability of these plants to both biotic and abiotic stresses. Numerous studies reveal that the accumulation of heavy metals in plant tissues can be used as an effective defense mechanism against stresses, especially biotic stresses, given that heavy metals can also be toxic to plant consumers. While trace metals and metalloids existed in the Earth’s crust originally, their concentrations in soil have escalated due to anthropogenic activities such as mining, irrigation, and the application of pesticides and fertilizers [6]. Among this range of metals and metalloids, plants absorbed both essential and non-essential elements. Essential trace metals like zinc (Zn), copper (Cu), iron (Fe), and cobalt (Co) actively participate in plant metabolism, but excessive concentrations can pose a risk by exceeding the plant tolerance threshold. On the other hand, non-essential metals such as arsenic (As), cadmium (Cd), chromium (Cr), mercury (Hg), and lead (Pb) can infiltrate plants through transporters meant for essential elements. Even at low concentrations, heavy metals pose a threat to plants by inducing the formation of reactive oxygen species (ROS), leading to oxidative damage across various components of the plant, including proteins, lipids, carbohydrates, and DNA [17]. When the concentrations of essential metals exceed the thresholds and non-essential metals accumulate in plants, it can result in phytotoxicity, leading to growth retardation, inhibition of leaf and root growth, reduced crop yield, and potential threats to food safety [18]. However, through long-term evolution, plants have developed mechanisms to utilize these harmful metals to resist external stress. Moreover, the development of innovative nanotechnology can assist metals and metalloids in enhancing nutrient management and mitigating the negative impacts of external stresses. Nanoparticles refer to nanoscale particles with dimensions less than 100 nm that have been applied in agriculture as nanofertilizers and nanopesticides [19]. Nanoparticles demonstrate higher dissolution rates compared to their bulk and ionic counterparts. These nanoparticles can be absorbed and distributed throughout the whole plant, signifying a potential breakthrough in agricultural revolution and food safety security through improving nutrient efficiency and pest prevention, reducing effects from adverse growth conditions [20,21,22]. Metal oxide nanoparticles (MeO-NPs), particularly those of silicon (Si), zinc (Zn), copper (Cu), titanium (Ti), and nickel (Ni), have received widespread attention for their role in enhancing plant defense ability [23]. Silicon and boron, both metalloids, have gained attention for their beneficial functions in reinforcing and stabilizing plants in challenging environmental conditions. However, it is important to note that this review does not encompass macrometal elements like calcium (Ca), potassium (K), and magnesium (Mg), along with trace metals such as iron (Fe), that are abundant in soil and hold crucial roles in plant metabolism activities.

The present study aims to comprehensively review recent research concerning the contributions of essential trace metals (Zn, Cu, Mn, and Ni), nonessential trace metals (Cd, Ti, and Ce), and beneficial metalloids (Si and B) in fortifying plant defense mechanisms against exogenous stressors. The goal is to explore the potential applications of these trace metals and metalloids in enhancing overall plant development.

## 2. Metal(Loid) Defense to Abiotic Stress

Under ongoing global climate change, abiotic stresses pose increasing challenges to plant growth, including issues like drought, salinity, and the adverse effect of heavy metal toxicity from harsh soil environments. Numerous studies have shown that plants can alleviate abiotic stresses through elemental adjustments. Metals and metalloids function in diverse ways, acting as both ions and nanoparticles, thereby playing a multifaceted role in improving plant resistance to abiotic stresses (Table 1).

### 2.1. Zinc (Zn)

Zn stands as an essential trace element crucial for the constitution of vital enzymes and structural domains within proteins, acting as a co-factor for many enzymes and crucial for protein folding [24]. However, when present in excess, Zn can become toxic for plants. Nevertheless, plants have evolved regulatory mechanisms to maintain Zn homeostasis, using it as an efficient tool to counter various stresses [25]. The strategic application of Zn has been observed to trigger plant defense mechanisms, thereby improving plant resistance against both abiotic and biotic stressors.

Zn operates within the ionic pathway to improve plant defenses, alleviating biotic stresses linked to heavy metal toxicity and various environmental stressors. Toxic elements like cadmium (Cd), known as one of the most toxic elements to plants, share similarities with Zn as both are divalent cations. Consequently, Cd can be uptaken and translocated through Zn transporters, such as OsZIP5 and OsZIP9 in rice, as well as AtHMA2 and AtHMA4 in *Arabidopsis thaliana* [26,27]. The presence of Zn holds a significant advantage in inhibiting Cd uptake, regulating the subcellular distribution of Cd, and stimulating the activities of antioxidative enzymes in plants [28]. These actions enhance the plant’s tolerance to Cd toxicity. Soil amendments involving Zn-lysine have shown significant potential for improving plant growth and enhancing enzyme activity related to the plant’s antioxidant response. Zn-lysine reduced Cd uptake, increased Zn contents in plants, and effectively counteracted the damaging effects caused by Cd [29]. Furthermore, apart from Zn ions, the utilization of zinc nanoparticles has proven effective in improving plant defense responses to Cd-induced stress. Wherein the Zn nanoparticles, ZnO NPs, showed remarkable applications in industry and agriculture [30]. ZnO-NPs contribute to enhancing plant tolerance, promoting plant growth, sustaining photosynthetic efficiency, triggering antioxidant responses, and reducing toxic metal accumulation [31]. Application of ZnO-NP as a foliar spray stimulates enzymatic activities and modulates the antioxidant system, effectively scavenging ROS and thereby promoting plant growth, which may fortify plant defense mechanisms against Cd stress. Notably, both foliar and soil applications of Zn have shown promising results in significantly reducing wheat grain Cd accumulation across different wheat cultivars, irrespective of their Cd accumulation levels, indicating the participation of ZnO-NPs in the detoxification of Cd [32,33]. In addition to its impact on Cd stress, the application of Zn has shown remarkable potential for alleviating toxicity as well. Utilizing ZnO-NPs has proven effective in improving soybean tolerance to As, reversing growth inhibition, and alleviating oxidative stress by participating in chloroplast composition. This approach also improves antioxidant enzyme activities, significantly enhancing plant antioxidant capabilities [34]. Moreover, ZnO-NPs have demonstrated the ability to regulate the opening and closing of bamboo stomata, contributing to the bamboo’s antioxidant capacity. Their application has also shown promising results in inhibiting As and Hg while reducing their translocation to aerial tissues [35]. In addition to toxic element stress, Zn application can also alleviate abiotic stress associated with environmental factors like salinity stress. Biofortification of Zn in maize has exhibited a positive impact on mitigating salinity effects, specifically associated with enhancing photosynthesis and counteracting the reduction in water and turgor potentials caused by salinity stress. Zn has also been observed to trigger plant defense responses to salinity stress without influencing the plant elemental content [36]. In instances where wheat faces the combined stresses of salinity, waterlogging, and drought, intrinsic Zn within plant seeds has demonstrated the capacity to limit oxidative damage while enhancing plant growth and nutrient absorption. Zn-lysine assisted with biochar has shown potential in countering the negative effects of salinity, thereby regulating plant physiological performance [37,38].

### 2.2. Silicon (Si)

Silicon is one of the pivotal metalloids that participate in plant growth. While traditionally considered a non-essential element, numerous studies have reported the potential beneficial function of Si to enhance plant strength, improve plant defense mechanisms, and mitigate both biotic and abiotic stresses [11,39]. Si can be absorbed by plants through polar-localized transporters (Lsi1 and Lsi2) in a passive manner. Insufficient accumulation of Si can lead to yield losses in rice production [39]. Si demonstrates a multitude of functions in fortifying plant defense mechanisms against exogenous stressors. Particularly noteworthy is the potential role of Si in ameliorating heavy metal toxicity through both external and internal pathways, a facet that has attracted increasing attention in research circles [40].

On the external aspect, SiO_2_ application in the soil decreases pH levels and decreases the mobility of heavy metals. Consequently, this action decreases the bioavailability of heavy metals in soil, which limits their uptake by plants [41]. Application of silica amendments to soil results in its absorption and accumulation within phytolith structures, subsequently reducing the accumulation of heavy metals such as Pb, Zn, Cd, Al, As, and Cu in plants [42,43]. When plants are exposed to Al-contaminated soil, exogenously applied Si binds to and strengthens the plant cell wall. This fortification promotes increased cell wall extensibilities, competes for binding sites, and reduces Al absorption. Additionally, Si ameliorates callose deposition induced by Al treatment, improving root border cell viability and forming a protective barrier against Al toxicity [44]. The interaction between Si and Al can foster the formation of Si-Al complexes on the border cells of the rice root surface, reducing Al transfer into plants [45]. Internally, the accumulation of Si in plants can alleviate heavy metal toxicity by affecting metal absorption and distribution, which is achieved by inducing the plant’s antioxidant response and regulating the expression of genes associated with both metal absorption and the plant’s defense response. Similarly, when facing Al toxicity exposure, silicon dioxide nanoparticles (SiNPs) have been observed to regulate enzyme activities linked to the antioxidant defense system. This regulation effectively ameliorates physiological disturbances and oxidative damage, supporting the enhancement of Al detoxification [46].

Si has the capacity to mitigate metal toxicity caused by the excessive accumulation of essential metals like manganese (Mn), nickel (Ni), and copper (Cu). Mn is crucial for plant metabolism; its excessive accumulation can lead to phytotoxicity, potentially affecting plant growth and development and even resulting in Mn toxicity in humans through the food chain [47,48]. In the case of Mn presence, Si has demonstrated its efficacy in rice roots by down-regulating the expression level of the Mn transporter *OsNramp5*. This regulatory action helps alleviate Mn toxicity by reducing Mn absorption. Si can also combine with Mn, forming Mn-Si complexes that decrease the translocation of Mn from roots to shoots [49]. Ni is another essential nutrient element that becomes toxic at supraoptimal concentrations. However, the exogenous application of Si has been proven to be beneficial in alleviating Ni toxicity by up-regulating the accumulation of antioxidant defense components in plants. This action induces a robust plant defense response, countering the negative effects caused by Ni toxicity [50]. A similar detoxification mechanism involving Si has been observed in mung bean seedlings experiencing Cu toxicity [51].

Si has garnered considerable attention for its role in detoxifying plants from toxic heavy metals and metalloids, especially in As detoxification. The high toxicity of both inorganic and organic As poses a threat to food safety, with As(III) being absorbed through Si transporters. Silicic acid (Si(OH)_4_) competes with As(III) for binding sites in the soil and also for the OsNIP2;1 transporter. Meanwhile, the application of SiO_2_ improves the competitiveness of Si(OH)_4_ [52]. Through Si application, enzyme activities such as ascorbate peroxidase (APX) and glutathione (GSH) reductase, involved in the ascorbate-glutathione cycle (AsA-GSH cycle), that are responsive to ROS scavenging, can be regulated [53]. Si regulation of the AsA-GSH cycle has shown promise in mitigating As(V) toxicity in maize [54]. Moreover, solubilized Si, assisted by *Bacillus amyloliquefaciens*, has demonstrated a significant reduction in As(III), As(V), and DMA content in rice [55]. Beyond its effects on As, Si application can alleviate the toxicity of heavy metals such as Cd, Cr, and Pb.

The application of Si can significantly improve various plant growth parameters, including plant biomass and chlorophyll content, while also inducing Si accumulation that triggers antioxidant defense mechanisms [56]. Si effects extend to the formation of suberin lamellae and deposition in casparian strip pores, thus establishing apoplasmic barriers that decline Cd availability in wheat and maize [57,58]. At the molecular level, SiNPs induce plant defense by enhancing the expression of antioxidative defense-related genes, consequently reducing Cd translocation and mitigating Cd toxicity in rapeseed [59]. Si has demonstrated its efficacy in alleviating Cr toxicity, encompassing both Cr(VI) and Cr(III). SiNPs application significantly decreases Cr uptake in seedlings and regulates the expression level of genes associated with photosynthesis, antioxidants, and plant defense mechanisms. SiNPs limit Cr distribution within plants by regulating GSH transcript levels, which assist in the sequestration of Cr in root vacuoles [60]. Hormones serve as pivotal signaling molecules involved in plant detoxification processes [3]. Si application not only increases the synthesis of GSH and auxin, which are responsible for developmental processes, but also upregulates antioxidant genes, induces root hair formation, and facilitates Cr(VI) detoxification in rice and *Brassica juncea* [61,62]. Si can obstruct Cr(III) transport capacity via the apoplastic route, with Cr(III) even potentially promoting Si polymerization in roots, which indicates the effectiveness of Si application in Cr(III) detoxification [63]. Moreover, Si can alleviate Pb toxicity through synergic interactions with fungi. The co-application of Si and *Bacillus subtilis* has shown a more effective decrease in Pb toxicity for *Solanum melongena*. Si application can significantly improve plant growth and reverse oxidant damage caused by Pb [64]. Based on the above advances, Si application can alleviate the toxicity of heavy metals and metalloids by reducing their accumulation in plants at ionic, metabolic, and molecular levels.

Apart from its role in heavy metal detoxification, the accumulation of Si within plants can also alleviate abiotic stresses caused by extreme environmental conditions, such as drought, salinity, and heat. In the presence of Si, root architecture undergoes reshaping through increased root endodermal silicification and suberization, subsequently increasing the root capacity for water uptake to effectively cope with drought stress [65]. Moreover, the application of Si-Zn combinations has shown significant promise in ameliorating drought stress by inducing antioxidant defense mechanisms and promoting the synthesis of proteins crucial for sustaining plant growth, thereby preventing production reduction [66]. The mechanism of the Si-induced plant defense responses is also evident in alleviating salt stress. It contributes to delaying leaf senescence by regulating redox processes, preserving ion homeostasis, and facilitating chlorophyll biosynthesis, ultimately enhancing plant defense abilities and attenuating the adverse effects of salinity and drought on plant photosynthesis [67,68]. When plants are exposed to various stresses, they tend to accumulate higher levels of ROS, which can damage biomolecules and affect metabolic activities. However, the application of Si can strengthen plant tolerance against exogenous abiotic stress by regulating the modification of secondary metabolites and enhancing plant resistance to ROS [12,69].

### 2.3. Boron (B)

As a trace metalloid, B participates in cell wall architecture and acts as an intermediary in sensing signals from the external environment and transmitting them internally [70,71]. Similar to other trace elements, deficiency in B can inhibit root elongation and lead to leaf chlorosis [72]. However, excessive amounts of B have shown a detrimental impact, exacerbating both biotic and abiotic stresses experienced by plants [73,74]. Nevertheless, previous research has indicated B’s ability to alleviate both biotic and abiotic stresses, especially under extreme conditions. B can mitigate various metal toxicities, notably Al. Its accumulation within the cell wall promotes increased cell wall thickness and callose formation and reduces the binding sites of Al^3+^ ions. This alteration affects the adsorption and desorption of Al^3+^ by reducing the demethylation of pectin, which is responsible for Al binding [71,75]. Additionally, the presence of B results in decreased expression of *Nrat1*, a gene responsible for coding an Al transporter, consequently reducing Al accumulation in plant roots [71]. B can also alleviate Cd toxicity in rice and *Brassica napus* by increasing Cd adsorption to cell walls and inducing plant antioxidant responses, ultimately leading to reduced Cd accumulation in plants [76,77]. The inclusion of B treatment has demonstrated the capacity to enhance watermelon tolerance to vanadium (V) stress. B induces the plant’s antioxidant defense mechanisms, boosting the chelation of V within the cell wall. At the molecular level, B facilitates the expression of genes responsible for V transport and sequestration into the vacuole of root cells, which efficiently reduces V translocation to aerial tissues [78]. B supplementation can relieve oxidative damage caused by abiotic stress. For instance, using pumpkin rootstock for grafting, known for its ability to absorb B, improved watermelon’s defense ability against oxidative stress [79]. When exposed to salt, drought, and extreme temperature stresses, plant photosynthesis and respiration tend to suffer. However, B accumulation can act as a protective shield, alleviating plant damage by balancing cation concentrations and maintaining cell wall elasticity [74]. When exposed to salt stress, supplementation with B decreases cellulose accumulation, protecting the cell wall and stabilizing its structure. The role of B in relieving oxidative stress is also notable; it diminishes malondialdehyde and H_2_O_2_ levels while enhancing components of the antioxidant defense system, such as the ascorbate/dehydroascorbate ratio, which tends to decrease under salt-induced stress [80,81].

### 2.4. Cerium (Ce)

Under abiotic stress conditions, plants often face the harmful accumulation of ROS, causing damage to crucial biomolecules like proteins, lipids, carbohydrates, and polynucleic acids [82]. Nanoceria (cerium oxide nanoparticles) can be used as a catalyst to increase the scavenging of ROS within chloroplasts both in vitro and in vivo, serving to protect plant photosynthesis [83,84]. When faced with salt stress, nanoscale CeO_2_ alterations modify root formation and influence Na^+^ fluxes, resulting in more Na^+^ translocation to shoots and less accumulation in roots. This adjustment efficiently maintains Na^+^/K^+^ homeostasis, ultimately improving plant tolerance to salt stress by 69.5% to 86.8%. Foliar-applied cerium oxide nanomaterials can change the transcript levels of genes related to antioxidative defense mechanisms, effectively countering oxidant stress. In addition, this application increases carbon source availability, enhancing rhizobacterial enrichment [85]. Nanoscale CeO_2_ can shorten root length to accelerate Na^+^ transport to aerial tissues, thereby protecting roots from the adverse effects of salt stress [86]. When faced with complicated stress combined with heavy metals and salt, the addition of nanoceria can trigger antioxidant defense systems and significantly relieve damage to rice seedlings [87].

### 2.5. Titanium (Ti)

Ti is beneficial for plant growth, and its application can promote plant development and boost crop yield, particularly when applied in low concentrations. In current applications, Ti primarily takes the form of nanoparticles within plants. Nano-titanium dioxide (TiO_2_ NPs) is extensively applied worldwide and acts as an antimicrobial agent, capable of triggering plant defense responses [88]. Numerous studies have revealed that the application of TiO_2_ NPs improves plant defense abilities against various abiotic stresses.

Application of TiO_2_ NPs exhibits a notable improvement in the plant’s detoxification abilities against various harmful heavy metals. The uptake of TiO_2_ NPs induces a robust antioxidant defense response in plants like sunflower and rice, significantly alleviating Cr(VI) and Cd toxicity by reducing their accumulation both in roots and shoots [89]. The application of TiO_2_ NPs has proven beneficial in alleviating negative effects caused by Cd and Cr(VI) stress, notably in preserving photosynthetic efficacy and bolstering antioxidant enzyme activities [90,91]. Moreover, TiO_2_ NPs can alleviate the inhibition of photosynthetic efficiency induced by Cr stress [91]. Both chemical and green TiO_2_ NPs significantly reduce As accumulation in mung beans while unveiling their toxicity to plants by stimulating the expression of genes associated with antioxidant mechanisms [92]. TiO_2_ NPs reduce Pb accumulation in rice plants by decreasing Pb bioavailability and regulating transcript levels of genes related to Pb transport. Simultaneously, TiO_2_ NPs improve plant growth by promoting nutrient utilization [93]. Furthermore, TiO_2_ NPs can alleviate plant damage and protect the photosynthetic system by boosting the activity of antioxidant enzymes and upregulating the expression of genes related to phytochelatins and glutathione synthesis. This action is notably potent when TiO_2_ NPs are combined with Si [94]. TiO_2_ NPs can alleviate the toxicity of heavy metals and metalloids by triggering plant antioxidant defense responses and reducing toxic element accumulation in plants, but the precise mechanism by which they achieve the detoxification response remains to be further investigated.

In addition to countering heavy metal stress, TiO_2_ NPs exhibit the potential to improve plant resistance against extreme environmental conditions, such as drought, salinity, and cold stress. The application of TiO_2_ NPs maintains leaf water content and decreases electrolyte leakage. These NPs stimulate nitrate reductase activity, which tends to be adversely affected by water deficit stress [95]. When applied, TiO_2_ NPs mitigate salt stress in marigolds by reducing oxidative damage to plants, thereby improving photosynthetic efficiency. Notably, TiO_2_ NPs not only enhance photosynthetic pigment contents but also nearly totally reverse the reduction of photosynthetic pigments caused by salinity stress. Additionally, their application increases antioxidant activity by 28.3%, compensating for the adverse effects of salinity stress [96]. The incorporation of TiO_2_ NPs into plants enhances the expression levels of genes related to plant antioxidative activities, aiding in combating cold stress [97]. In terms of plant growth, TiO_2_ contributes positively and assists multi-walled carbon nanotubes to reduce hydrogen peroxide content by nearly half, thereby mitigating heat stress-induced damage [98].

**Table 1 plants-13-00313-t001:** Metals and metalloids involved in plant defense against abiotic stress.

Metal(loid)	Species	Alleviation	Process	Reference
Zinc (Zn)	*Cosmos bipinnatus*	Cd toxicity	Inhibit Cd uptake, regulate Cd subcellular distributions, and induce the activities of plant antioxidative enzymes;	[28]
	*Triticum aestivum* L. and *Oryza sativa* L.	Cd toxicity	Improve plant growth and enhance enzyme activity related to the plant’s antioxidant response;	[29]
	*Lactuca sativa* L.	Cd toxicity	Improve plant growth, maintain photosynthesis efficiency, induce plant antioxidant responses, and reduce Cd accumulation;	[31]
	*Glycine max* L.	As toxicity	Reverse growth inhibition, alleviate oxidative stress;	[34]
	*Pleioblastus pygmaeus (Miq.) Nakai*	As and Pb toxicity	Increase plant antioxidant activities, regulate stomatal closure, inhibit toxic metal uptake, and regulate their translocation;	[35]
	*Zea mays* L.	Salinity stress	Alleviate the adverse effect of salinity associated with photosynthesis and reverse the reduction of water and turgor potentials caused by salinity stress;	[36]
	*Triticum aestivum* L.	Salinity stress	Inhibit the negative effect of salinity, improve plant growth, and regulate physiological performance;	[37]
	*Oryza sativa* L.	Cr(VI) toxicity	Increase GSH and auxin synthesis and induce root hair formation;	[61]
	*Oryza sativa* L.	Cr(III) toxicity	Obstruct Cr(III) transport capacity via the apoplastic route;	[63]
	*Solanum melongena*	Pb toxicity	Induce plant antioxidant activity, promote plant growth, and reverse oxidant damage caused by Pb;	[64]
	*Oryza sativa* L.	As(III) toxicity	Compete binding sites and transporters;	[52]
	*Zea mays* L.	As(V) toxicity	Regulate the AsA-GSH cycle and counterbalance ROS content;	[54]
	*Triticum aestivum* L.	Drought	Reshape root architecture and induce plant defense;	[66]
	*Brassica juncea*	Drought, salinity	Induce plant defense and maintain chlorophyll biosynthesis;	[67,68]
Boron (B)	*Glycine max* L.	Salinity stress	Enhance antioxidant defense and mitigate oxidative damage;	[80]
	*Gossypium* spp.	Salinity stress	Reverse the damage effect and protect cell wall structure and antioxidant defense-related components;	[81]
	*Poncirus trifoliate* (L.) *Raf.*	Al toxicity	Reduce the biding domain and demethylation of pectin in Al; down-regulate genes for Al transport;	[71,75]
	*Oryza sativa* L., *Pisum sativum*, and *Brassica napus*	Cd toxicity	Increase Cd adsorption to cell walls and induce plant antioxidant responses;	[76,77]
	*Citrullus lanatus*	V toxicity	Sequester V in the root cell vacuole, increase the chelation of V in the cell wall, and induce the plant’s antioxidant defense;	[78]
Cerium (Ce)	*Arabidopsis thaliana*	ROS damage	Act as a catalyzer used by chloroplast to increase the ROS scavenging;	[84]
	*Zea mays* L.	Salinity stress	Modified root formation, altered Na^+^ fluxes, plant defense response, and enhanced rhizobacterial enrichment;	[85]
	*Brassica napus*	Salinity stress	Shorten root length to accelerate Na^+^ transport to aerial tissues to relieve root stress;	[86]
	*Oryza sativa* L.	Cd and salt	Trigger antioxidant defense systems and significantly relieve damage;	[87]
Titanium (Ti)	*Oryza sativa* L.	Cd toxicity	Reduce Cd accumulation and alleviate negative effects caused by Cd stress;	[89,90]
	*Helianthus annuus* L.	Cr(VI) toxicity	Reduce Cr accumulation, preserve the photosynthetic pigments, and bolster antioxidant enzyme activities;	[91]
	*Vigna radiata* L.	As toxicity	Reduce As accumulation and induce the expression of genes associated with antioxidant mechanisms;	[92]
	*Oryza sativa* L.	Pb toxicity	Decrease Pb bioavailability and regulate the transcript level of genes related to Pb transport; improve plant growth by promoting nutrient utilization;	[93]
	*Vicia faba* L.	Water deficit	Maintain the water content of leaves, decrease electrolyte leakage, and induce nitrate reductase activity;	[95]
	*Calendula officinalis* L.	Salinity stress	Reduce oxidant damage, improve photosynthetic efficiency, and increase antioxidant activity;	[96]
	*Cicer arietinum* L.	Cold stress	Enhance the expression level of genes related to plant antioxidative activity;	[97]
	*Sesamum indicum* L.	Heat stress	Improve plant growth, reduce hydrogen peroxide content	[98]

## 3. Metal(Loid) Defense against Biotic Stress

Plants confront biotic threats, including herbivores and microbial invasions from viruses, bacteria, and fungi. Metals and metalloids play important roles in bolstering plant defenses against biotic stress. Elements like Cu and Ti can be toxic to plants at excessive concentrations, but when nanoparticles form, they enhance plant defense responses. In response to complex stress scenarios, plants have evolved strategies to use toxic metals to defend against attacks by biotic fungi and herbivores. Conversely, toxic metals such as Ni and Cd can be utilized as defenses against biotic stresses when present in low concentrations, following plant adaptation to their toxicity [99]. Metal(loid)s absorbed by plants from the soil accumulate in herbivores through the food chain. The presence of metal(loid)s in the soil-plant-herbivore-predator biosystem serves as a defense against biotic stresses, especially against herbivores, by inducing the emission of volatile organic compounds (VOCs) by plants, attracting predators of herbivores, and deterring attacks [100]. Numerous studies have revealed that jasmonic acid (JA) and salicylic acid (SA) play essential roles in enabling plant defense against biotic stresses that could be regulated by metals and metalloids accumulated in plants.

Among angiosperms, the proportion of metal hyperaccumulators stands at 0.2%, with a significant proportion found within the Brassicaceae family [101]. These hyperaccumulators demonstrate efficient evolution by increasing tolerance and converting the stress of harmful ions into efficient defenses against biotic attacks. This adaptation indicates the co-evolutionary relationship between plants and various stressors [102]. Hyperaccumulators employ two primary defense mechanisms: inorganic and organic defenses. Inorganic defense is achieved by direct poisoning, leveraging their exceptional ability to accumulate heavy metals, especially in aerial tissues. This accumulation acts as a deterrent, rendering these tissues toxic to herbivores that ingest them. On the other hand, organic defense mainly relies on plant chemical defense through the emission of volatiles [103]. Numerous studies showcase the utilization of hyperaccumulators, with a high accumulation capacity of heavy metals, in defending against pathogens and herbivores (Table 2).

### 3.1. Cadmium (Cd)

Cd stands out as one of the most toxic metals in soil, posing a potential risk not only to plants but also to human health. During plant evolution, especially for those thriving in contaminated soil, plants have developed detoxification strategies to withstand harsh soil environments. Hyperaccumulators such as *Thlaspi caerulescens* exhibit an exceptional capacity to accumulate heavy metals like Cd and Zn, proving to be efficient defenders against external stressors, especially pathogen invasion. In this context, Cd and Zn found in *Thlaspi caerulescens* significantly inhibit the growth of *Pseudomonas syringae* by 35% and 40%, respectively [104]. Another notable hyperaccumulator, *Brassica juncea*, not only serves as a major oil seed crop with high economic value but also exhibits Cd hyperaccumulation traits, making it a feasible candidate for phytoremediation and pathogen resistance purposes [105]. However, during its growth period, *B. juncea* can be infected by *Alternaria brassicicola*, causing Alternaria leaf blight, which leads to an economic reduction of 47% [106]. Under in vitro conditions, Cd exhibits the ability to completely inhibit *Alternaria brassicicola* growth at concentrations of 80 μM. When Cd accumulates in plants, it triggers the accumulation of JA and SA in leaves and induces the expression of pathogen attack-related genes and miRNAs. This results in bolstering the elemental defense of *B. juncea* against *Alternaria* infection [107]. In addition to hyperaccumulators, non-hyperaccumulator plants like invasive plants like *Ageratina adenophora* and *Alternanthera philoxeroides* have been documented to accumulate various heavy metals, including Cd, Cr, and Pb. Cd accumulation specifically serves as an ideal defense mechanism against pathogen damage, shedding light on how elemental defense helps invasive plants adapt to new ecological environments, indicating that higher Cd accumulation assists plant defense compared with native plants [108,109]. Laboratory experiments indicated that Cd significantly inhibits the growth of *Rhizoctonia solani* in vitro. When tested in vivo, Cd enhanced the resistance of the invasive plant *Ageratina adenophora* to *Rhizoctonia solani*, resulting in a reduction of lesions by 60% [108]. Compared to native plants, invasive species showed greater accumulation ability and tolerance to heavy metals and subsequently used Cd accumulation as a defense to resist herbivores. The enhancement of defense responses against herbivores due to Cd contamination underscores the mutual promotion observed between Cd contamination and biological invasion in the realm of invasion biology.

The elemental defense theory is also feasible for woody plants. Notably, *Populus yunnanensis* stands out as an optimal candidate for phytoremediation due to its high biomass yield and high trace element accumulation [110]. Meanwhile, the variation in the genotype of *Populus yunnanensis* significantly impacts the accumulation of trace metals (Zn, Cd, and Mg) in leaves. Interestingly, these metals can return to the soil during leaf abscission. Particularly, Cd accumulation in *Populus yunnanensis* leads to a restructuring of phyllosphere microorganisms, resulting in an improvement in resistance against *Pestalotiopsis microspore*. This exposure to Cd results in a notable reduction in lesion area on leaves, exhibiting a decrease of 41.6% and 35.0% in both male and female poplars, respectively [111].

Herbivores can face poisoning due to the accumulation of metals in plants. Leaves damaged by insect herbivores have been observed to accumulate more Cd compared to undamaged leaves. Consequently, the increased accumulation of Cd in leaves resulted in a remarkable 49% reduction in larvae consumption [112]. This finding indicates the defensive role of Cd in *A. halleri* against herbivores. Within experiments involving Brassicaceae specialists and generalist herbivores, paired choice experiments showed that both Zn and Cd can decrease the area of herbivore feeding. Furthermore, the combination of Zn and Cd exhibited an additional suppressive effect on herbivores, showcasing the defense mechanism of hyperaccumulator defense in herbivores [113]. Moreover, tomato plants exposed to Cd showed more tolerance to spider mites. When spider mites fed on leaves containing Cd, they showed a lower growth rate. These instances indicate that Cd exhibits toxicity to herbivores and inhibits their growth [114]. When present in plant phloem sap due to uptake from soil, Cd serves as an efficient defensive mechanism against aphids, which feed by sucking sap from the plant [115].

The accumulation of Cd in *Populus yunnanensis* leaves has a notable influence on herbivore lesions. Leaves with higher Cd accumulation exhibit protective effects against both specialist herbivores like *Botyodes diniasalis* and generalist herbivores like *Spodoptera exigua* [116]. Exposure of plants to Cd significantly induces the emission of VOCs, effectively deterring females of both specialist and generalist herbivore species from oviposition [117]. Meanwhile, the application of exogenous spermidine and nitrogen demonstrates a positive correlation between plant growth and the inhibition of Cd on herbivore growth [118,119]. Herbivores consuming leaves with higher Cd content show inhibited growth [116]. For instance, gypsy moth larvae feeding on leaves with high Cd accumulation showed a body weight that was only 32.9% of those fed control leaves, with almost all larvae succumbing within a week. In larch plants, Cd-induced stress prominently improves antioxidant enzyme activity and enhances resistance to gypsy moth larvae without affecting plant growth or biomass [120]. Transfer of Cd through a soil-plant-whitefly-ladybird beetle food chain revealed that Cd present in the soil can be absorbed by cotton plants, subsequently transferring to herbivorous whiteflies, and eventually accumulating in their predators. Notably, the accumulation of Cd in herbivores is significantly higher than in predators. This accumulation can reduce adult longevity, oviposition days, fertility, and the total pre-oviposition duration [100].

### 3.2. Zinc (Zn)

Zn is an essential metal that plays important roles in catalytic functions and protein structure within plants, but its concentration varies among plant species. *Arabidopsis halleri* was reported as a hyperaccumulator that could accumulate Zn and Cd at extremely high levels. Notably, when compared to its non-accumulator sister species, *Arabidopsis lyrate*, *A. helleri* showed a 76-fold increase in Zn accumulation and an 8-fold increase in Cd accumulation [101]. Zn accumulation in plants can induce plant defense ability to stressors. In instances where leaves were infected with *Alternaria brassicae* conidia, plants grown in soil enriched with high Zn concentrations showed an induced antifungal defense mechanism. This response included elevated camalexin levels in leaves, recognized for its characteristics in impeding pathogen growth [121,122]. Studies involving *Arabidopsis thaliana* plants with loss-of-function mutations in Zn transporters showed increased susceptibility to fungal infection [123]. Furthermore, the application of Zn has shown promise in improving photosynthesis and modifying defense-associated enzyme activities, thereby enhancing the defense response of tomato plants against early blight disease caused by *Alternaria solani*. Particularly noteworthy is the combination of Zn application with NPK fertilizer, which effectively reduces disease intensity [124].

ZnO-NPs have garnered significant attention due to their potential in antimicrobial applications [125]. Their application demonstrates an improvement in tobacco tolerance against wildfire caused by *Pseudomonas syringae* pv. *Tabaci*. ZnO-NPs exert their effect by impeding bacterial growth, including disruptions in physiological and metabolic events. Furthermore, these nanoparticles regulate the transcript levels of genes related to hormone signaling pathways, stimulating the closure of stomata to protect tobacco from bacterial invasion [126]. In both in vitro and in vivo conditions, ZnO-NPs exhibit an inhibitory impact on the growth of bacterial *Acidovorax citrulli*. Their application not only improves plant growth but also activates plant antioxidant responses against bacterial infections, thereby restricting the incidence of bacterial fruit blotch in melons [127]. Remarkably, in rice plants, ZnO-NPs offer complete protection against rice blast disease. Their action includes the destruction of the *Magnaporthe grisea* cell wall, leading to the rupture of hyphae. Moreover, the adverse effects caused by *Magnaporthe grisea* inoculation can be reversed by the application of ZnO-NPs. Intriguingly, plants treated with ZnO-NPs even showed superior growth parameters compared to plants without fungal infection [128]. When applied to soil, ZnO-NPs act as nanofertilizers, facilitating enhanced plant growth parameters. Additionally, they serve as nanofungicides with equivalent disease suppression to antifungal agents [129].

### 3.3. Copper (Cu)

Cu is an essential trace element that participates in multiple fundamental biological processes within plants. It plays an indispensable role in critical plant functions like photosynthesis, respiration, and ethylene perception. However, when present in excess, it can become toxic for plants. Over time, plants have developed mechanisms to regulate Cu levels to maintain homeostasis and have used it as an antimicrobial agent. This is evident in the usage of various Cu-based fungicides, such as bordeaux mixtures, employed to prevent fungal and bacterial diseases. Previous studies have demonstrated that Cu plays an independent role in bolstering plant immunity, showcasing its significance as a crucial component in pesticides and fertilizers.

Cu-containing nanopesticides present a promising solution for addressing the limitations observed in bulk and ionic pesticides concerning delivery and utilization [130]. Exposure of plants to Cu(OH)_2_ nanopesticides initiates changes in the transcript levels of antioxidants and detoxification enzymes, thereby stimulating plant defense mechanisms. Notably, both nano and bulk forms of Cu(OH)_2_ reinforce the presence of ascorbate peroxidase (APX) and trigger plant defense mechanisms in *Zea mays* seedlings [131]. Furthermore, exposure to Cu(OH)_2_ nanopesticides leads to changes in the transcript levels of antioxidant and detoxification-related genes, resulting in the improvement of plant resistance against abiotic and biotic stresses [132]. CuO nanoparticles demonstrate significant efficacy in reducing Fusarium wilt fungus and simultaneously enhancing the growth and yield of tomato plants. The addition of CuO nanoparticles results in an increase in plant fresh weight of 64%. Interestingly, these nanoparticles induce an antifungal response in plants rather than directly acting on the fungus [133]. In a different context, the application of CuSO_4_-based copper-chitosan nanoparticles (CuChNp) proved beneficial in enhancing the resistance of millet plants against *Pyricularia grisea*. This application improves defense enzyme activities, inhibiting blast disease development and concurrently enhancing plant growth and yield [134].

Apart from fungi, Cu-based bactericides are becoming increasingly popular in crop protection. Studies examining the antibacterial properties of nanocopper against bacterial blight in *Punica granatum* have demonstrated its ability to inhibit bacterial activity at extremely low concentrations in vitro. Additionally, nanocopper exhibits a preventive effect by impeding bacterial cells from colonizing plant tissues [135]. Furthermore, the optimal concentration of CuO nanoparticles proves effective in inhibiting the growth of *Cnaphalocrocis medinalis* without negatively affecting rice growth [136].

### 3.4. Silicon (Si)

Si can assist plants in resisting exogenous biotic stresses. Numerous instances have reported the involvement of Si/SiO_2_NPs in bolstering plant defenses against biotic stresses. This involvement includes improving plant tolerance through physical reinforcement, the synthesis of secondary compounds, and the alteration of molecular expression [137]. These findings suggest promising and sustainable prospects for Si-based applications in agriculture [138].

The significance of Si in enhancing plant resistance against biotic stress has been well documented. Si plays an essential role in mediating the interaction between plants and pathogens, actively engaging in plant defense responses against fungal and bacterial infections across physiological, biochemical, and molecular levels [137,139,140]. Externally, the application of Si to the soil affects soil microbial communities within the rhizosphere. Si modulates the microbial structure and hampers microbial growth, thereby shaping the intricate microbial ecosystem and regulating plant nutrient absorption [141]. Moreover, Si cooperates with Cu^2+^ to improve the efficiency of bactericides, effectively inhibiting the growth of pathogens. The incorporation of organosilicon in bactericides not only reduces disease index but also allows for a decrease in bactericide concentration, paving the way for more environmentally friendly disease control methods [142].

The application of Si enhances plant defense mechanisms against prokaryotic pathogens. Introducing silicate-based materials and silica nanoparticles to tomato plants increases Si content in both shoots and roots. Si accumulation, which promotes increased antioxidant activity, particularly when plants face infection from *Ralstonia solanacearum*, plays a crucial role in neutralizing ROS at the biochemical level. Additionally, Si inhibits the expression of genes related to the regulation of bacterial virulence, thereby reducing bacterial pathogenicity [143]. Notably, the addition of Si reduces wilt severity by 62.5% compared to soil lacking Si supplementation. When Arabidopsis plants encounter *Pseudomonas syringae* infection, SiO_2_NPs induce systemic acquired resistance (SAR), including signaling compounds related to plant pathogenesis [138].

Si application enhances plant defense against eukaryotic microorganisms. Si and SiO_2_NPs strengthen the plant’s resistance ability by attaching pectin and hemicellulose, thereby fortifying the cell wall. Si also participates in cell wall modification and biosynthesis [44]. Its impact on fungal conidia is noteworthy. The application of Si significantly suppresses the growth and germination of *Colletotrichum* sp. Conidia in vitro and effectively hampers anthracnose development on chili peppers [144]. Moreover, Si application offers protection to soybeans against *Phytophthora sojae* invasion by depositing beneath the cell wall, forming physical barriers in the roots, and mechanically strengthening the plant. Plants treated with Si significantly prevent the interaction between *P. sojae* and the plant by interrupting the release of effectors. These plants exhibit a decline in transcriptomic response induced by *P. sojae* infection, in contrast to plants without Si application [145]. Si maintains plant mentalism and alleviates the negative effects of fungal infection on plants. It ameliorates various parameters affected by plant fungal infection, such as the photosynthetic rate, hormone signaling, and ROS accumulation. Si application notably enhances photosynthetic performance, thereby boosting plant resistance to *Puccinia graminis* [146]. In rice, Si enhances resistance to *Magnaporthe oryzae* infection by stimulating plant immunity through hormone signaling, particularly SA, which triggers a plant resistance response to suppress fungus growth and conidia germination [147]. Si also improves plant resistance at the transcript level by inducing the expression of genes related to the antioxidant response and mitigating pathogen-induced transcript changes. This attribute explains the Si capacity for mitigating *Plasmodiophora brassicae* invasion in Brassica crops [148]. Si-treated tomato plants exhibit improved gene transcript levels related to JA and SA, which are responsible for plant defense activity. Si also increases antioxidative enzyme activity, improving plant resistance to early blight caused by *Alternaria solani* [149].

In addition to its antimicrobial effects, Si accumulation can efficiently enhance plant defense mechanisms against herbivores. Enhanced Si deposition in the apoplast triggers architectural changes in plant morphology upon herbivory attacks. This Si accumulation within plants reduces palatability and inflicts damage upon herbivore mouths, thereby deterring herbivore feeding and minimizing damage [150]. Exposure to Si significantly amplifies Si-hyperaccumulating grass damage against chewing herbivores, providing immediate and heightened defense mechanisms. Si absorption within plants effectively reduces herbivore damage to plants, synchronizing with the wear of herbivore mandibles [151]. Furthermore, the inclusion of Si amendments in plant stems significantly induces Si transport into plants, increasing their mechanical strength. This increase in strength shortens the length of feeding tunnels created by the pink stem borer, subsequently protecting plants from herbivores [152].

At the physiological level, the addition of Si promotes various plant organ activities, including structural architecture, photosynthetic efficiency, and enzyme synthesis [153]. When plants face attacks from arthropod pests, the synergic action between Si and JA induces the accumulation of secondary metabolites known as herbivore-induced plant volatiles (HIPV). These compounds act as key indicators of insect herbivory and function to attract natural enemies of herbivores, such as predators and parasitoids [154,155]. Notably, the emission of HIPV can attract the predator *Helicoverpa armigera* to *Dicranolaius bellulus* and the predator *Manduca sexta* to *Geocoris* spp., subsequently deterring pest attacks on plants [156,157]. At the molecular level, Si can upregulate defense genes to enhance plant resistance against insect attacks effectively [153]. The introduction of Si amendments also induces the transcript levels of genes responsible for Si transport and plant defense regulation. This augmentation contributes to enhancing plant defense responses, particularly against stem borer [152]. Notably, Si-mediated increases in the transcript of genes related to JA synthesis and perception accelerate JA accumulation following the herbivore attack [158].

### 3.5. Nickel (Ni)

Ni is one of the essential nutrients in plants, acting as a component of metalloenzymes. However, like Zn and Cu, excessive concentrations of Ni can prove toxic to plants. When rice cultured in paddy soil was treated with nickel-chitosan nanoparticles (Ni-Ch NPs), there was a 3-fold reduction in disease severity compared to plants cultured with no-Ni-Ch NPs when confronted with *Pyricularia oryzae* infection. Furthermore, Ni accumulation in hyperaccumulator species like *Thlaspi caerulescens* significantly inhibits the growth of *Pseudomonas syringae*, exhibiting a notable 35% reduction [104]. Ni-Ch NPs efficiently stimulate plant growth, enhancing seed germination, shoot development, and proliferation of lateral roots, which enhances plant defenses against pathogens [159]. Feeding experiments focusing on herbivores consuming various plant parts, including leaves, roots, xylem, and phloem, revealed that Ni hyperaccumulation in Brassicaceae plants significantly reduces the survival rates of herbivores feeding on leaves and vascular tissues while interrupting the growth of whitefly populations [160]. Notably, *Spodoptera exigua* larvae feeding on Ni-hyperaccumulated plant organs showed a mortality rate of 66%, indicating the toxic effects of Ni on the larvae [161].

### 3.6. Titanium (Ti)

TiO_2_ NPs also play an important role in plant resistance to biotic stresses. Their application has proven effective in enhancing plant resistance against pathogens, including bacteria, fungi, and viruses. Primarily, TiO_2_ NPs can mitigate both bean yellow mosaic virus and broad bean stain virus effects on faba bean plants [162,163]. Observations reveal that TiO_2_ NPs remarkably limit the interaction between viruses and plant cells, fostering improved plant growth characteristics such as enhanced photosynthetic activity and water content. Moreover, their application upregulates the accumulation of defense-related proteins, leading to a reduction in the severity of viral infections. Exogenous application of TiO_2_ NPs upregulates proteins related to plant defense mechanisms and secondary metabolism. This action improves wheat resistance to yellow rust disease and alleviates the morbidity caused by *Puccinia striiformis* in leaves. TiO_2_ NPs also increase the transcript levels of phytoalexin-deficient genes, which can modulate phytoalexin synthesis. This augmentation contributes to improving plant defenses against the pathogens, elucidating the beneficial effects of TiO_2_ NPs in aiding Arabidopsis plants to cope with *Botrytis cinerea* infections [164].

### 3.7. Other Metals

Boron (B) accumulation also amplifies plant resistance against biotic stressors. Fertilization with B improves the resistance of silver birch trees against autumnal moths. Female moths consuming birch leaves from plants with added B exhibit a significant reduction in pupal weight and immunity, showcasing the impact of boron fertilization on insect resistance [165]. Gadolinium (Gd), a rare earth element known for its biostimulant properties, exhibits remarkable effects. The application of Gd improves root elongation in plants. Notably, when *Arabidopsis thaliana* plants are infected with *Botrytis cinerea*, the addition of Gd induces a robust plant defense response at transcript levels against fungal infection. This response significantly reduces the proportion of lesions on leaves [166]. In the case of the invasive plant *Phytolacca americana* treated with high concentrations of Mn, there is a notable enhancement in the inhibitory effect on herbivores, leading to a 66% reduction in herbivore weight [99].

**Table 2 plants-13-00313-t002:** Metals and metalloids involved in plant defense against biotic stress.

Metal(loid)	Species	Alleviation	Process	Reference
Cadmium (Cd)	*Thlaspi caerulescens*	*Pseudomonas syringae*	Inhibit the growth of *Pseudomonas syringae* and enhance plant disease resistance;	[104]
	*Brassica juncea*	*Alternaria brassicicola*	Enhance JA and SA accumulation in leaves and induce the expression of pathogen-related genes;	[107]
	*Ageratina adenophora*	*Rhizoctonia solani*	Inhibit the growth of pathogens and reduce lesion aeras;	[108]
	*Populus yunnanensis*	*Pestalotiopsis microspora*	Restructure the consistency of the phyllosphere microorganisms; improve plant resistance to pathogens;	[111]
	*Arabidopsis halleri*	*Pieris rapae*	Reduce the insect herbivory feeding rate;	[112]
	*Populus yunnanensis*	*Botyodes diniasalis* and *Spodoptera exigua*	Decrease the generalist herbivore survival rate; reduce the specialist herbivore feeding rate;	[113]
	*Solanum lycopersicum*	*Tetranychus urticae*	Toxic to herbivores and inhibit herbivore growth;	[114]
	*Arabidopsis halleri*	*Myzus persicae*	Present in phloem and can be toxic to aphids;	[115]
	*Populus yunnanensis*	*Stauronematus compressicornis* and *Plagiodera versicolora*	Induce volatile organic compounds; prevent the oviposition of females of both specialist and generalist herbivores;	[117]
	*Larix olgensis*	*Lymantria dispar*	Improve antioxidant enzyme activity and enhance resistance to gypsy moth larvae; reduce body weight of larvae; improve;	[120]
Zinc (Zn)	*Arabidopsis halleri*	*Alternaria brassicae*	Induce antifungal defense and promote the synthesis of camalexin;	[122]
	*Nicotiana attenuata*	*Pseudomonas syringae*	Interrupt the growth of bacteria, regulate the transcript level of genes related to the hormone signal pathway, and stimulate the closeness of stomata;	[126]
	*Cucumis melo*	*Acidovorax citrulli*	Improve plant growth, activate the plant’s antioxidant response to bacterial infection, and restrict the incidence of bacteria;	[127]
	*Oryza sativa* L.	*Magnaporthe grisea*	Destruct the cell wall of *Magnaporthe grisea*; reverse the negative effect caused by fungi;	[128]
	*Brassica juncea*	*Alternaria brassicae*	Facilitate plant growth and suppress disease occurrence;	[129]
Copper (Cu)	*Lycopersicon esculentum*	*Fusarium oxysporum*	Reduce Fusarium wilt fungus estimates and increase the growth and yield of tomato plants;	[133]
	*Eleusine* *coracana* Gaertn.	*Pyricularia grisea*	Improve defense enzyme activities and inhibit the development of blast disease;	[134]
	*Oryza sativa* L.	*Cnaphalocrocis medinalis*	Inhibit *Cnaphalocrocis medinalis* growth;	[136]
Silicon (Si)	*Saccharum* spp. hybrids	*Xanthomonas albilineans*	Reduce the disease index and simultaneously lower the bactericide concentration;	[142]
	*Solanum lycopersicum*	*Ralstonia solanacearum*	Stimulate antioxidant activity, eliminate ROS, and inhibit gene expression levels related to the regulation of bacterial virulence to reduce bacterial pathogenicity;	[143]
	*Arabidopsis thaliana*	*Pseudomonas syringae*	Induce systemic, acquired resistance;	[138]
	*Glycine max*	*Phytophthora sojae*	Deposit under the cell wall, then form physical barriers and mechanically strengthen the plant, preventing interaction between *P. sojae* and the plant;	[145]
	*Capsicum annuum* L.	*Colletotrichum* sp.	Suppress the growth and germination of *Colletotrichum* sp. conidia;	[144]
	*Avena sativa* L.	*Puccinia graminis* f. sp. *avenae*	Ameliorate photosynthetic parameters affected by fungus and enhance plant resistance;	[146]
	*Oryza sativa* L.	*Magnaporthe grisea*	Stimulate plant immunity, establish plant resistance responses;	[147]
	*Brassica napus*	*Plasmodiophora brassicae*	Reverse the transcript-level changes caused by the pathogen and induce transcript-level changes related to plant antioxidant activity;	[148]
	*Lycopersicon esculentum* Mill.	*Alternaria solani*	Improve gene transcript levels related to JA and SA; increase antioxidative enzyme activity;	[149]
	*Brachypodium distachyon*	*Helicoverpa armigera*	Reduce herbivore damage, synchronized with the mandible wear of herbivores;	[151]
	*Eleusine coracana* Gaertn.	*Sesamiainferens* Walker.	Increase the mechanical strength of plants and shorten the pink stem borer feeding tunnel length;	[152]
	*Oryza sativa* L.	*Pyricularia oryzae*	Enhance plant growth and enhance plant defense to pathogens;	[159]
Nickel (Ni)	*Thlaspi caerulescens*	*Pseudomonas syringae*	Inhibit the growth of *Pseudomonas syringae;*	[104]
	*Streptanthus polygaloides*	*Tetranychus urticae*	Reduce the survival of herbivores and interrupt the growth of whiteflies;	[160]
	*Vicia faba* L. Fabaceae	*Bean yellow mosaic virus*	Reduce disease severity and increase plant growth indices; induce plant antioxidant responses;	[163]
Titanium (Ti)	*Vicia faba* L. Fabaceae	*broad bean stain virus*	Reduce disease severity and induce the expression of genes involved in the SA signaling pathway;	[162]
	*Arabidopsis thaliana*	*Botrytis cinerea*	Increase the transcript level of the phytoalexin-deficient gene and improve plant defense;	[164]
	*Phytolacca americana*	*Spodoptera litura*	Enhance plant defense against herbivore; inhibit herbivore growth;	[99]
Manganese (Mn)	*Betula pendula* Roth	*Epirrita autumnata*	Improve the resistance of silver birch to autumnal moths and reduce pupal weight and immunity;	[165]
Boron (B)	*Arabidopsis thaliana*	*Botrytis cinerea*	Improve root elongation, induce plant defense, and reduce the proportion of lesions in leaves	[166]

## 4. Conclusions and Future Perspectives

Plants, as sessile organisms, navigate a complex array of environmental conditions in the soil and aerial atmosphere passively. Over time, plants have evolved multiple mechanisms to combat exogenous stresses. Plants uptake essential nutrient elements from the soil to maintain their life cycle and production. The nutrient elements that are absorbed and accumulated in plants are crucial for agricultural productivity and, subsequently, considerable for human nutrition. Element accumulations have emerged as integral players in plant defense, giving rise to the element defense hypothesis (EDH). Preliminary studies predominantly support the EDH in hyperaccumulate plants, indicating the positive function of metals and metalloids in plant defense abilities to exogenous stressors. Subsequent studies revealed that, besides hyperaccumulators, trace metals in non-hyperaccumulating plants can also improve plant resistance to external stressors.

This review consolidates recent decade-focused processes, highlighting the involvement of metals and metalloids in plant defense mechanisms. The accumulation of these elements holds promising potential to enhance plant resistance against both biotic and abiotic stresses. Notably, metal(loid) nanoparticles exhibit extraordinary applications in agriculture, particularly owing to the transformative capabilities afforded by nanotechnology. This advancement enhances the efficiency of metal(loid) utilization in agricultural practices, presenting promising solutions to various challenges in the field. The absorption and accumulation of nanoparticles within plants offer multifaceted avenues for stress mitigation. In precision farming, the use of nanoparticles for safe agricultural cultivation has shown diverse and beneficial effects. Nanoparticles, identified as phytomedicine, play a compelling role in combating anti-microbial resistance [23]. The impressive advantage of NPs is their large surface area coupled with their high sorption capacity. This characteristic becomes especially advantageous in nanofertilization, exemplified by ZnO-NPs, which can significantly boost plant growth while reducing nutrient release and improving fertilizer use efficiency [167].

Within plants, these metals and metalloids play a pivotal role in defense mechanisms and alleviating stress, operating through external, physiological, biochemical, and molecular pathways (Figure 1). Metal(loid) presence has demonstrated pivotal roles in both external and internal aspects. Externally, their presence can regulate the physical and chemical properties of soil, depositing on root surfaces and reducing the mobility and availability of toxic metals, thereby alleviating abiotic stress. Internal mechanisms can be divided into three categories: (1) At the physiological level, metals and metalloids like silicon strengthen plants by binding to cell walls and competing for toxic metal binding sites. Metal(loid) accumulation triggers regulation of the cell wall and stomata modification in response to stress. (2) At the biochemical level, metals and metalloids trigger vital plant defense responses, including antioxidant defense mechanisms and secondary metabolite modifications that alleviate stress-induced inhibition. The adverse damage of ROS to plants can be alleviated as well. Moreover, heavy metals, while potentially detrimental, can paradoxically bolster plant defenses against biotic herbivores through their high accumulation in plants. (3) At the molecular level, metal(loid) accumulation regulates the expression levels of genes associated with plant growth, metal absorption, and defense responses. This regulation subsequently enhances plant resistance against exogenous stresses. The present study provides a prospect for the application of metals in metalloids in agriculture to improve plant resilience against a spectrum of biotic and abiotic stresses. However, from advanced processing, metals and metalloids do participate in plant defense responses from various aspects; the identification of plant defense is still based on the physiological phenotypes, and the precise mechanism under the plant defense response still needs further investigation.

## Figures and Tables

**Figure 1 plants-13-00313-f001:**
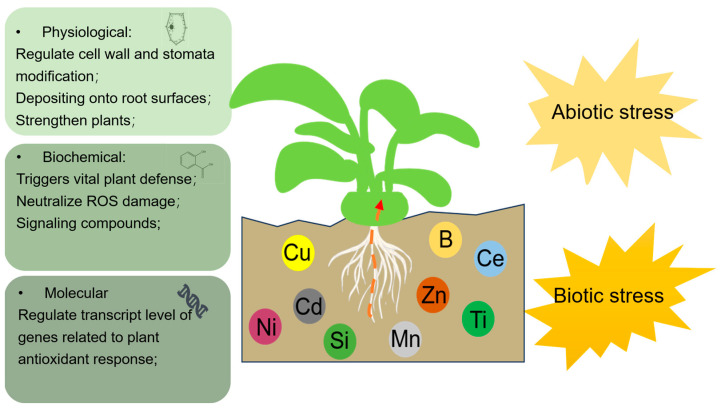
Schematic of the involvement of metals and metalloids in plant response to biotic and abiotic stress.

## Data Availability

Not applicable.

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
