# Peer review of "Advances in the Involvement of Metals and Metalloids in Plant Defense Response to External Stress"

_plants, 2024, doi:10.3390/plants13020313_

Round 1

Reviewer 1 Report

Comments and Suggestions for Authors

This review article describes metal(liod) defense to abiotic and biotic stress.  It surveys a number of metal(liod)s such as Zn, Si, b, Ce, Ti, Cd, Cu, and Ni and in so doing, it also mentions others like Pb, Hg, As.  However, throughout the review article, there is a strong concentration on nanoparticles.  While this is a great addition to the review article, it is not mentioned in the abstract. Anyone interested in nanoparticles and their aid in plant defense mechanisms would not know to look at this review article.  Please add content to the abstract about the nanoparticle discussion within the review article.  In addition, a comment on the nanoparticles should be mentioned in the Conclusion and future perspectives. 

Comments on the Quality of English Language

The English is quite well done except for the abstract.  There are a few areas of improvement needed in the abstract. 

Author Response

Referee #1

This review article describes metal(liod) defense to abiotic and biotic stress.  It surveys a number of metal(liod)s such as Zn, Si, b, Ce, Ti, Cd, Cu, and Ni and in so doing, it also mentions others like Pb, Hg, As.  However, throughout the review article, there is a strong concentration on nanoparticles.  While this is a great addition to the review article, it is not mentioned in the abstract. Anyone interested in nanoparticles and their aid in plant defense mechanisms would not know to look at this review article.  Please add content to the abstract about the nanoparticle discussion within the review article. In addition, a comment on the nanoparticles should be mentioned in the Conclusion and future perspectives.

Answer: We are very grateful for the advises, nanoparticles do play effective roles in enhancing plant defense abilities, extensive research progresses showed their function in alleviating external stress. We have added description about nanoparticles relevance and prospect for application in the abstract and conclusion, respectively. English has been further polished as reviewer advised.

Please see the attachment with MS revised as suggested.

Reviewer 2 Report

Comments and Suggestions for Authors

Dear Authors

I read with great interest the presented review “Advances in involvement of metals and metalloids in plant defense mechanism”. In this review, the authors try to substantiate the previously stated hypothesis, according to which plants uptake elements as a defense mechanism against pathogen attack and herbivory consumption [9].” In contrast to this hypothesis, the authors of the review extended it not only to biotic, but also to abiotic stress and, it seems to me, prematurely called it a theory.

In general, despite the relevance of the topic of analysis, the review is purely phenomenological in nature, although the authors made an attempt to some generalize the presented material in the final part of the manuscript and in the schematic drawing that is given at the end of the introduction, but it seems to me that it would be more logical to place it in end of the review. In addition, this figure does not contain any elements indicating the specific action of metals and metalloids under biotic and abiotic stress.

While overall positive about the review, I would like to make some comments.

1. There is no novelty in the authors discussing the protective role of essential elements as cofactors of important metabolic enzymes. This was described tens and tens of years ago. When discussing the protective role of such elements, it would be nice to identify mechanisms that are not associated with the implementation of their function as cofactors of macromolecules.

2. From my point of view, the protective role of cadmium under biotic stress, the biological function of which in plants has not yet been determined, is of particular interest. In this regard, it would be desirable to pay special attention to the description of the known mechanisms of the protective effect of cadmium.

3. It would be nice to present the protective effect of metals (metalloids) under biotic stress in the form of a separate generalizing scheme, paying special attention to the mechanism of their action.

4. It would be interesting to discuss the question of why the most effective plant protection by metals occurs in nanoform, and not in the form of free metals or their ions.

5. The protection of plants from biological factors, including animals, is often associated with the ability of plants to synthesize tens and hundreds of thousands of secondary compounds, many of which exhibit biological activity. It is advisable to reflect this point at the beginning of the review. It also cannot be excluded that the protective effect of metals on plants in many cases may be mediated by the result of their stimulation of the synthesis of certain secondary compounds.

6. The authors often associate the protective effect of some metals with the ability of certain plant species to accumulate increased levels of toxic elements. This circumstance makes the authors' recommendation to use metals or metalloids to improve the resistance of important crops somewhat strange.Dear Editor

Kind regards

Round 2

Reviewer 2 Report

Comments and Suggestions for Authors

Dear Authors

I believe that the authors were attentive to the comments and wishes expressed and took most of them into account. The text of the manuscript has been significantly corrected. Overall the manuscript has been improved.

Taking into account the above, I recommend accepting the manuscript for publication as submitted.

Kind regards